# Efficacy of a Third-Generation High-Vision Ultrathin Endoscope for Evaluating Gastric Atrophy and Intestinal Metaplasia in *Helicobacter pylori*-Eradicated Patients

**DOI:** 10.3390/jcm11082198

**Published:** 2022-04-14

**Authors:** Junichi Uematsu, Mitsushige Sugimoto, Mariko Hamada, Eri Iwata, Ryota Niikura, Naoyoshi Nagata, Masakatsu Fukuzawa, Takao Itoi, Takashi Kawai

**Affiliations:** 1Department of Gastroenterological Endoscopy, Tokyo Medical University Hospital, Shinjuku, Tokyo 160-0023, Japan; juju1113u@yahoo.co.jp (J.U.); mahamada0820@gmail.com (M.H.); eiwata@tokyo-med.ac.jp (E.I.); rniikuratritonocn@gmail.com (R.N.); nnagata_ncgm@yahoo.co.jp (N.N.); t-kawai@tokyo-med.ac.jp (T.K.); 2Department of Gastroenterology and Hepatology, Tokyo Medical University Hospital, Shinjuku, Tokyo 160-0023, Japan; masakatu8055@yahoo.co.jp (M.F.); itoitakao@gmail.com (T.I.)

**Keywords:** third-generation ultrathin endoscope, transnasal endoscopy, narrow-band imaging, Kyoto classification of gastritis, gastric atrophy and intestinal metaplasia

## Abstract

Background: Image-enhanced endoscopy methods such as narrow-band imaging (NBI) are advantageous over white-light imaging (WLI) for detecting gastric atrophy, intestinal metaplasia, and cancer. Although new third-generation high-vision ultrathin endoscopes improve image quality and resolution over second-generation endoscopes, it is unclear whether the former also enhances color differences surrounding atrophy and intestinal metaplasia for endoscopic detection. We compared the efficacy of a new third-generation ultrathin endoscope and an older second-generation endoscope. Methods: We enrolled 50 *Helicobacter pylori*-eradicated patients who underwent transnasal endoscopy with a second-generation and third-generation endoscope (GIF-290N and GIF-1200N, respectively) in our retrospective study. Color differences based on the International Commission on Illumination 1976 (L*, a*, b*) color space were compared between second-generation and third-generation high-vision endoscopes. Results: Color differences surrounding atrophy produced by NBI on the GIF-1200N endoscope were significantly greater than those on GIF-290N (19.2 ± 8.5 vs. 14.4 ± 6.2, *p* = 0.001). In contrast, color differences surrounding intestinal metaplasia using both WLI and NBI were similar on GIF-1200N and GIF-290N endoscopes. NBI was advantageous over WLI for detecting intestinal metaplasia on both endoscopes. Conclusions: NBI using a third-generation ultrathin endoscope produced significantly greater color differences surrounding atrophy and intestinal metaplasia in *H. pylori*-eradicated patients compared with WLI.

## 1. Introduction

In Japan, the Kyoto classification of gastritis was developed to identify patients at high risk of developing gastric cancer using a grading system for endoscopic risk that is based on endoscopic characteristics of gastritis related to *Helicobacter pylori* infection [1,2]. This system assesses patients for gastric cancer risk by scoring five endoscopic parameters, namely atrophy, intestinal metaplasia, enlarged folds, nodularity, and diffuse redness [1,2]. Given that gastric cancer arising from long-term *H. pylori* infection is a major concern around the world, and *H. pylori*-positive patients develop gastric cancer at a rate of 0.4% annually [3], it is important to accurately evaluate the endoscopic risk of gastric cancer, especially in terms of gastric atrophy and intestinal metaplasia, in *H. pylori*-positive and previously eradicated patients at high risk of gastric cancer [2,4,5].

Recent developments in endoscopic instrumentation and image-enhancement techniques, known together as image-enhanced endoscopy (IEE), including narrow-band imaging (NBI), blue laser imaging, linked color imaging, and texture and color enhancement imaging with or without magnification, have improved the detection rate of gastric atrophy, intestinal metaplasia, and cancer [6,7,8,9,10]. Although transnasal and oral endoscopes differ in their detection rates for gastrointestinal diseases and the assessment of the severity of gastritis [11,12], in 2020, Olympus Co. developed a third-generation high-vision ultrathin endoscope called GIF-1200N with a new high-quality complementary metal-oxide semiconductor sensor that could improve resolution, noise, and graduation over older second-generation endoscopes such as GIF-290N. In addition, the development of a new processor called EVIS X-1 has improved image quality over older processors such as EXERA III and LUCERA ELITE. Therefore, combining the new third-generation ultrathin endoscope with the new processor is expected to increase the detection rate of endoscopic atrophy, intestinal metaplasia, and cancer over a combination of the older second-generation endoscope and older processor [13]. Although determining color differences according to the International Commission on Illumination (CIE) 1976 (L*a*b*) color space is a typical objective endoscopic analysis method [10,14,15], it is unclear whether third-generation endoscopes produce significantly greater color differences surrounding atrophy and intestinal metaplasia for endoscopic detection.

In Japan, transnasal endoscopy using an ultrathin endoscope has become a popular medical screening test because it is relatively pain-free for patients. Given their widespread use, it is prudent to evaluate the usefulness of these tests. Here, we investigated whether a new high-vision third-generation ultrathin endoscope using white-light imaging (WLI) and NBI improves the detection of gastric cancer risk assessed based on the Kyoto classification of gastritis compared with an older second-generation endoscope in *H. pylori*-eradicated patients.

## 2. Materials and Methods

### 2.1. Study Design and Patients

This study was a retrospective trial conducted at Tokyo Medical University Hospital to investigate the efficacy of transnasal endoscopy for evaluating the severity of gastritis in patients who had received eradication therapy for *H. pylori* infection. Of patients who had a history of transnasal endoscopy with a second-generation endoscope (GIF-290N, Olympus Co., Tokyo, Japan), we enrolled 50 patients aged ≥ 20 years who had subsequently undergone endoscopy with a high-vision third-generation ultrathin endoscope as part of a health check-up (GIF-1200N, Olympus Co., Tokyo, Japan). Exclusion criteria were patients with a history of gastric surgery, a lack of clear images to evaluate endoscopic gastritis, atrophy and intestinal metaplasia, and no history of eradication therapy for *H. pylori* infection.

The study protocol adhered to the ethical principles of the Declaration of Helsinki and was approved by the institutional review board of Tokyo Medical University (T2020-0059). Because this study was conducted under a retrospective design, and written informed consent was not obtained from each enrolled patient, a document describing an opt-out policy through which potential patients and/or relatives could refuse inclusion was uploaded on the Tokyo Medical University Hospital website.

### 2.2. Endoscopy and Severity of Gastritis

Endoscopy was performed using the older second-generation GIF-290N (from March 2015 to June 2020) and new third-generation ultrathin GIF-1200N endoscopes (from April 2021 to January 2022) (Figure 1). The severity of gastritis was retrospectively scored according to the Kyoto classification of gastritis and the Kimura–Takemoto classification [1,16]. In the Kyoto classification, the total score is calculated by summing the scores for five parameters, namely atrophy, intestinal metaplasia, hypertrophy of gastric folds, nodularity, and diffuse redness [2,17,18,19]. The 1st and 2nd endoscopies were performed by an expert endoscopist (KT). Two expert endoscopists independently evaluated the severity of gastritis using WBI. When scores assigned by the two endoscopists differed, a consensus was reached by reviewing patient images.

### 2.3. Measurement of Colors

Color differences surrounding atrophic borders and intestinal metaplasia were measured and compared between GIF-290N and GIF-1200N endoscopes [10]. We randomly set three pairs (with and without) of regions (atrophy or intestinal metaplasia) of interest and calculated the color difference in each patient using pictures of similar anatomical location (i.e., antrum and lesser curve of lower body) at both GIF-290N and GIF-1200N endoscopes. Color differences were calculated using the CIE 1976 (L*, a*, b*) color space [20,21], a three-dimensional model composed of a black-white axis (L*, brightness), a red-green axis (a*, red-green component), and a yellow-blue axis (b*, yellow-blue component). The color difference was defined as ΔE, which expresses the distance between two points in the color space. ΔE was calculated using the following formula: {(ΔL*)^2^+ (Δa*)^2^+(Δb*)^2^}^1/2^, where ΔL*, Δa*, and Δb* are differences in the L*, a*, and b* values, respectively, between regions with and without atrophy and intestinal metaplasia. Each ΔL*, Δa*, and Δb* value was determined by a computer operator who was blinded to clinical information using Adobe Photoshop, version 22.5.1 (Adobe KK, Tokyo, Japan).

### 2.4. Statistical Analysis

Parameters including age, height, body weight, and Kyoto classification score are expressed as mean ± standard deviation (SD). Categorical variables for GIF-290N and GIF-1200N endoscopes are summarized as n (%) and were compared using χ^2^ tests. Statistically significant differences in mean Kyoto classification scores and mean ΔE between GIF-290N and GIF-1200N endoscopes were determined using Student’s *t*-test. A *p*-value < 0.05 was considered statistically significant, and all *p*-values were two-sided. All statistical analyses were performed using the statistical analysis software SPSS, version 27.0 (IBM Japan, Tokyo, Japan).

## 3. Results

### 3.1. Patient Characteristics

Of patients who had a history of endoscopy using GIF-290N, we randomly enrolled 50 *H. pylori*-eradicated patients, who had subsequently undergone transnasal endoscopy using the GIF-1200N endoscope as part of a health check-up. The mean age was 74.3 ± 6.6 years and 54.0% of patients were male (Table 1). Baseline diseases included peptic ulcers in 10.0% of patients (*n* = 5), gastric cancer in 6.0% (*n* = 3), and other cancers in 24.0% (*n* = 12). Drugs taken included proton pump inhibitors in 32.0% of patients (*n* = 16), antiplatelet drugs in 32.0% (*n* = 16), and anticoagulants in 8.0% (*n* = 4) (Table 1). The mean duration from 1st to 2nd endoscopies was 35.5 ± 19.8 (months ± SD).

### 3.2. Severity of Gastritis Assessed Using WLI on GIF-290N and GIF-1200N Endoscopes

On WLI, the severity of gastritis based on the degree of atrophy scored according to the Kimura–Takemoto classification, and the degree of atrophy, intestinal metaplasia, enlarged folds, nodular gastritis, and diffuse redness scored according to the Kyoto classification of gastritis were similar between GIF-290N and GIF-1200N endoscopes (Table 2). The rates of detection of xanthoma, multiple white and flat elevated lesions and map-like redness were also similar between GIF-290N and GIF-1200N endoscopes (data not shown).

Similarly, mean endoscopic scores of gastritis evaluated using WLI according to the Kyoto classification of gastritis were not significantly different between GIF-290N and GIF-1200N endoscopes (Table 3).

### 3.3. Color Differences in Endoscopic Atrophy and Intestinal Metaplasia on GIF-290N and GIF-1200N Endoscopes

The color difference surrounding atrophic borders evaluated using NBI was 14.4 ± 6.2 on the GIF-290N endoscope and 19.2 ± 8.5 on the GIF-1200N endoscope; the difference was significantly greater on the GIF-1200N than the GIF-290N endoscope (*p* = 0.001, Table 4). Significant differences evaluated using NBI were also observed along the black–white (ΔL*) and the red–green axis (Δa*) surrounding atrophy between GIF-290N and GIF-1200N endoscopes. In contrast, there was no significant color difference surrounding atrophic borders observed using WLI between the two endoscopes (Table 4).

Likewise, there were no significant differences in ΔE surrounding intestinal metaplasia using either WLI (7.1 ± 3.3 and 6.9 ± 3.1) or NBI (13.7 ± 4.23 and 13.3 ± 4.4) on the GIF-290N compared to the GIF-1200N endoscope (Table 4).

When color differences surrounding atrophic borders were compared between WLI and NBI on the GIF-1200N endoscope, the difference in NBI (19.2 ± 8.5) was significantly greater than that in WLI (13.4 ± 6.4, *p* < 0.001, Figure 2). In contrast, no significant differences were observed in ΔE surrounding atrophic borders between WLI and NBI on the GIF-290N endoscope. The color differences surrounding intestinal metaplasia using NBI on the GIF-290N and GIF-1200N endoscopes were 13.3 ± 4.4 and 13.7 ± 4.2, which were significantly greater than those observed using WLI (6.9 ± 3.1 and 7.1 ± 3.3, both *p* < 0.001, respectively) (Figure 2).

## 4. Discussion

We demonstrated that NBI with a third-generation ultrathin endoscope produced significantly greater color differences based on the CIE 1976 (L*, a*, b*) color space surrounding atrophic borders than a second-generation endoscope in *H. pylori*-eradicated patients. In addition, NBI using a third-generation endoscope produced significantly greater color differences surrounding atrophy and intestinal metaplasia compared with WLI. Therefore, combining third-generation endoscopes with IEE methods such as NBI may be useful for identifying patients at a high risk of gastric cancer at health check-ups through not only improved image quality, resolution, noise, and graduation, but also increased color differences.

### 4.1. Efficacy of Third-Generation High-Vision Ultrathin Endoscope for Detecting Gastric Cancer Risk

Given that advances in endoscopic technology have markedly enhanced the diagnostic capability of endoscopy, it is important to determine the best method for identifying patients with a higher risk of gastric cancer and the best endoscopic method for diagnosing early-stage gastric cancer in *H. pylori*-positive and *H. pylori*-eradicated patients. The detection rate of gastric cancer and diagnostic efficiency for gastritis among those at a high risk of developing gastric cancer is affected by the skill of the endoscopist (e.g., experience and knowledge concerning endoscopy and gastric cancer), gastric environment (e.g., *H. pylori* infection, mucus, mucosal redness, and the severity of atrophy and intestinal metaplasia), tumor-related factors (e.g., location, size, form, number of tumors, and pathological types), and endoscope image-related factors (e.g., image quality, resolution, noise, graduation, type of IEE, field of view and ease of passage).

Because transnasal endoscopy performed using an ultrathin endoscope is safe and can be performed without any sedation, endoscopic examination is often performed transnasally to reduce invasiveness and distress to the patient [22,23,24]. However, major disadvantages of transnasal endoscopy include the need for complex considerations (e.g., anesthesia of the nasal cavity, use of vasoconstrictors, limited manipulations, and lower power aspiration and air supply), poor image quality, lower detection rate of gastrointestinal diseases, and different evaluation methods for the severity of gastritis compared with oral endoscopy [12,25]. In fact, when we investigated the capability of a second-generation endoscope for diagnosing gastric cancer in 255 consecutive patients who underwent gastrointestinal screening, the sensitivity, specificity, and diagnostic accuracy for gastric cancer diagnosis using WLI were as low as 50.0%, 63.6%, and 61.5%, respectively [11]. Likewise, Toyoizumi et al. [12] reported that the sensitivity and specificity of a second-generation ultrathin endoscope for diagnosing gastric cancer were significantly lower than that of a high-resolution oral endoscope (sensitivity 58.5% vs 78%, *p* = 0.021; specificity 91.8% vs 100%, *p* = 0.014). However, the introduction of a third-generation ultrathin endoscope (GIF-1200N; Olympus Co., Tokyo, Japan) with improved resolution in 2020 has made it possible to obtain high-definition images of the microsurface of the mucosa. Although few studies have reported the usefulness of this third-generation ultrathin endoscope with high image quality for detecting gastric cancer, our previous preliminary study showed that the sensitivity of WLI with the third-generation GIF-1200N endoscope was 85.7% for gastric cancer diagnosis, which is a significant improvement on the second-generation GIF-XP290N endoscope (31.8%) [13]. Clearly, distinguishable colors at the border of a tumor, atrophy and intestinal metaplasia can enhance an endoscopist’s ability to determine the extent of the tumor, atrophy, and intestinal metaplasia. Therefore, in addition to endoscope image-related factors such as image quality, resolution, noise, and graduation, assessing color differences according to the CIE 1976 (L*a*b*) color space is also an objective endoscopic analysis method (e.g., detection of intestinal metaplasia and atrophic border between WLI and IEE) [10,14,15]. In this study, we demonstrated for the first time that NBI using a third-generation ultrathin endoscope produces significantly greater color differences surrounding atrophic borders than a second-generation endoscope. The high image quality, high resolution, and low noise of the third-generation ultrathin endoscope are expected to improve the detection rate of gastric cancer, and screening for gastric cancer using third-generation ultrathin endoscopes will become increasingly important as the number of patients who have received *H. pylori* eradication therapy increases.

### 4.2. NBI Using Ultrathin Endoscopy for Identifying Patients at High Risk of Gastric Cancer

Gastric cancer is relatively common worldwide, with over 1,000,000 new cases reported in 2018, and an estimated 783,000 reported deaths [26]. A recent meta-analysis showed that eradication therapy reduces the risk of gastric cancer, with a relative risk of 0.51–0.67 [27], although it cannot prevent gastric cancer development in all patients. Therefore, the surveillance of all *H. pylori*-eradicated patients is important for the early diagnosis of gastric cancer. In fact, regular endoscopic surveillance increases the survival rate of gastric cancer, with a Japanese cohort study showing that >95% of cases with gastric cancer identified by annual endoscopy surveillance can be cured by endoscopic resection [28].

The current *H. pylori* infection rate in Japan is about 35%, and most Japanese patients with atrophy undergoing a health check-up by endoscopy have previously received eradication therapy for *H. pylori* infection. Because endoscopic characteristics differ between *H. pylori*-positive and eradicated patients, we focused on strategies for detecting atrophy and intestinal metaplasia in *H. pylori*-eradicated patients in this study. Although WLI is currently the most common endoscopic technique for the stomach, a diagnosis by WLI is based on the size and color of the lesion, and the characteristics of the lesion surface, surrounding mucosa, and gastric rugae. This makes the experience of the endoscopist immensely important and causes great variability in diagnostic findings. Despite sufficient illumination by a new processor for endoscopy, abnormalities in mucosal discoloration and morphological changes to the mucosal surface go undetected using WLI or a combination of WLI and indigocarmine chromoendoscopy in 18.9–21.6% of early-stage gastric cancer cases [29,30]. A recent multicenter randomized controlled trial also failed to show the efficacy of second-generation NBI for early-stage gastric cancer detection in high-risk patients (1.9% in WLI and 2.3% in second-generation NBI, *p* = 0.412) [31]. However, recent studies have reported that the vessel plus surface (VS) classification system, which uses innovative optical image-enhancing technology in magnifying NBI endoscopy to enable the better visualization of surface structures and blood vessels than WLI, has high diagnostic capability for determining the extent of invasion of early-stage gastric cancer and superficial squamous cell carcinoma of the head and neck and the esophagus [29,30,32,33]. In fact, the MAPS II guideline, an official statement from the European Society of Gastrointestinal Endoscopy, states that using a high-definition endoscope with IEE is more effective for detecting atrophy and intestinal metaplasia than high-definition WLI alone [34]. Similarly, guidelines for the endoscopic diagnosis of early gastric cancer from the Japan Gastroenterological Endoscopy Society Guideline Committee [5] recommend using conventional WLI for determining the depth of invasion of early gastric cancer (Strength of recommendation: 2, Level of evidence: C) and IEE for diagnosing the extent of invasion (Strength of recommendation: 1, Level of evidence: B). A randomized controlled trial by Asada-Hirayama et al. reported that magnifying NBI endoscopy produced significantly better results than indigocarmine chromoendoscopy in endoscopic submucosal dissection (ESD) cases (89.4% vs. 75.9%, *p* = 0.007) [29]. Additionally, we previously reported that the diagnostic accuracy of a second-generation ultrathin endoscope using WLI is as low as 61.5% but can be increased to 92.3% by adding observation with NBI [11]. In this study, color differences surrounding atrophic borders and intestinal metaplasia were significantly greater on a third-generation ultrathin endoscope using NBI (19.2 ± 8.5 and 13.7 ± 4.2, *p* < 0.001) than WLI (13.4 ± 6.4 and 7.1 ± 3.3, both *p* < 0.001). Our findings provide the first evidence of the efficacy of NBI with a third-generation ultrathin endoscope for detecting gastric atrophy and intestinal metaplasia as indicators of precancerous lesions over WLI. Large-scale multi-center prospective studies are needed to investigate the efficacy of NBI with a third-generation endoscopy for detecting atrophy, intestinal metaplasia, and gastric cancer, and for identifying those at high risk among not only *H. pylori*-eradicated patients, but also *H. pylori*-positive and naive patients.

### 4.3. Limitations

This study has a few limitations. First, it was a retrospective single-center study and the sample size was small. Second, endoscopic evaluations of gastritis using GIF-290N and GIF-1200N were performed at different times. In general, gastric atrophy and intestinal metaplasia after eradication therapy in any patients are gradually recovered depended on age, severity of atrophy, and intestinal metaplasia at eradication, and environmental and genetic factors [35,36]. In this study, because the mean duration from first to second endoscopies was 35.5 ± 19.8 months, the severity of gastritis may differ between GIF-290N and GIF-1200N endoscopes. Because two endoscopies should be carried out at the same time to compare ability, this study may have potential biases. However, mean endoscopic scores of gastritis evaluated using WLI according to the Kyoto classification of gastritis were not significantly different between GIF-290N and GIF-1200N endoscopes, of which the difference in gastric mucosal situation by the duration from first to second endoscopies may ignore. However, we will plan to clarify the clinical endoscopic significance between GIF-290N and GIF-1200N at the same time, as part of further study. Third, we evaluated the severity of gastritis by combining GIF-1200N with the EVIS X-1 processor and GIF-290N with the LUCERA ELITE processor; therefore, differences in processor functionality may have affected the results. Fourth, although pathological examination is considered the gold standard for the evaluation of gastric atrophy and intestinal metaplasia, we did not evaluate pathological gastritis in this study. Therefore, we cannot completely deny a hypothesis that the apparently better image quality of a particular imaging technique for an evaluation of atrophy and intestinal metaplasia may actually be an artifact during the image management process.

## 5. Conclusions

We showed that NBI with a third-generation high-vision ultrathin endoscope enhanced color differences surrounding atrophy and intestinal metaplasia according to the CIE 1976 (L*a*b*) color space compared to a second-generation endoscope in *H. pylori*-eradicated patients. Thus, third-generation ultrathin endoscopes may possess greater diagnostic efficiency for screening and improve risk stratification for gastric cancer.

## Figures and Tables

**Figure 1 jcm-11-02198-f001:**
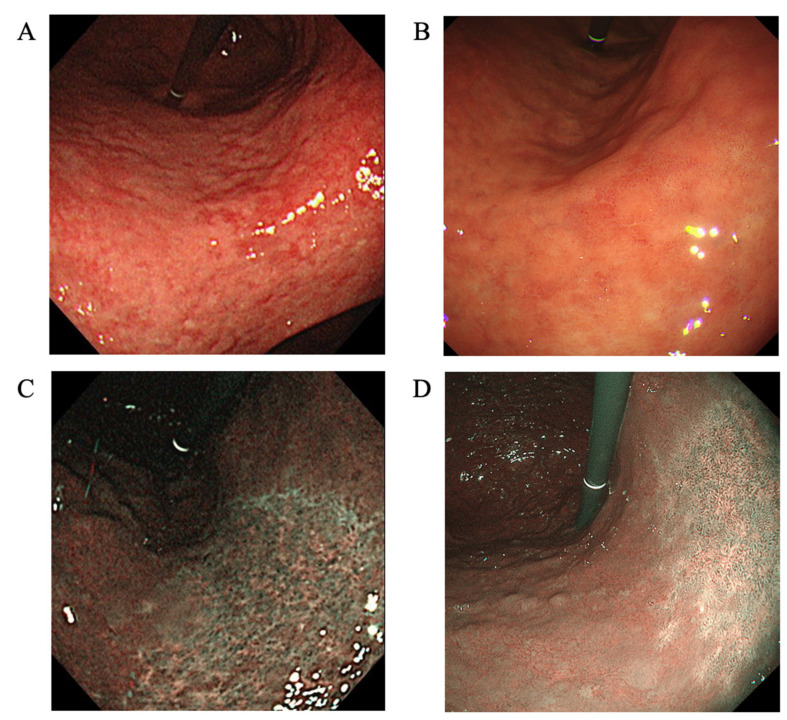
Images taken using a second-generation GIF-290N ultrathin endoscope by white-light imaging (**A**), third-generation GIF-1200N ultrathin endoscope by white-light imaging (**B**), second-generation endoscope by narrow-band imaging (**C**), and third-generation endoscope by narrow-band imaging (**D**). The endoscopic features of mucosal atrophy are characterized by a discolored mucosa and visible capillary network in the atrophic area (**A**–**D**). Intestinal metaplasia is defined as multiple ashen nodular or cobblestone-like lesions on atrophic mucosa observed (**A**,**B**). Villous appearance, whitish mucosa, and rough mucosal surface are helpful indicators for the endoscopic diagnosis of intestinal metaplasia. Map-like redness is defined as reddish depressed areas of various shape and sizes in the atrophic area (**A**).

**Figure 2 jcm-11-02198-f002:**
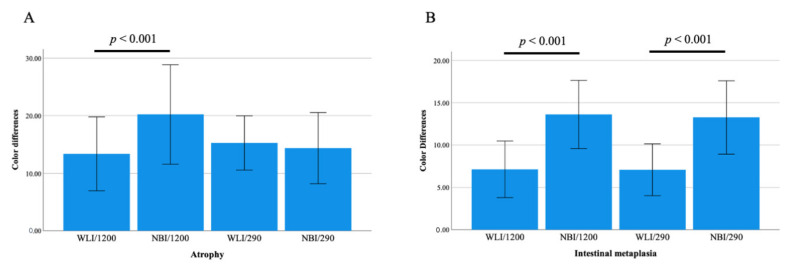
Color differences surrounding atrophic borders (**A**) and intestinal metaplasia (**B**) between WLI and NBI using third-generation GIF-1200N and second-generation GIF-290N ultrathin endoscopes.

**Table 1 jcm-11-02198-t001:** Characteristics of patients enrolled in this study.

	All Patients (*n* = 50)
Age (years ± SD)	74.3 ± 6.6
Sex [male, *n* (%)]	27 (54.0%)
Height (cm ± SD)	160.8 ± 8.7
Body weight (kg ± SD)	58.8 ± 9.7
*H. pylori* infection, Negative/Current/Eradicated [*n*/*n*/*n*]	0/0/50
Smoking, Never/Current/Past [*n*/*n*/*n*]	32/1/17
Alcohol [*n* (%)]	32 (64.0%)
Diseases	
Peptic ulcer [*n* (%)]	5 (10.0%)
Gastric cancer [*n* (%)]	3 (6.0%)
Cancer (others) [*n* (%)]	12 (24.0%)
Hyperlipidemia [*n* (%)]	17 (34.0%)
Hypertension [*n* (%)]	28 (56.0%)
Diabetes mellitus [*n* (%)]	7 (14.0%)
Drugs	
Proton pump inhibitor [*n* (%)]	16 (32.0%)
NSAID [*n* (%)]	0 (0%)
Antihypertensive drug [*n* (%)]	32 (64.0%)
Antihyperlipidemic drug [*n* (%)]	20 (40.0%)
Antiplatelet drug [*n* (%)]	16 (32.0%)
Anticoagulant [*n* (%)]	4 (8.0%)
Antidiabetic drug [*n* (%)]	9 (18.0%)
Bisphosphonate [*n* (%)]	2 (4.0%)
Mean duration from 1st to 2nd endoscopies (months ± SD)	35.5 ± 19.8

Abbreviations: *H. pylori*: *Helicobacter pylori*, NSAID: non-steroidal anti-inflammatory drug, SD: standard deviation.

**Table 2 jcm-11-02198-t002:** Severity of gastritis by white-light imaging according to the Kimura–Takemoto classification and Kyoto classification of gastritis.

Category/Characteristic		GIF-1200N Endoscope (WLI)	GIF-290N Endoscope (WLI)	*p*-Value
Kimura–Takemoto classification				
Atrophy	C-O–C-II	9 (18.0%)	8 (16.0%)	0.240
	C-III–O-I	29 (58.0%)	36 (72.0%)	
	O-II–O-III	12 (24.0%)	6 (12.0%)	
Kyoto classification of gastritis				
Atrophy	A0	2 (4.0%)	0 (0%)	0.420
	A1	13 (26.0%)	16 (16.7%)	
	A2	35 (70.0%)	34 (66.7%)	
Intestinal metaplasia	IM0	15 (30.0%)	11 (22.0%)	0.662
	IM1	12 (24.0%)	15 (30.0%)	
	IM2	23 (46.0%)	24 (48.0%)	
Enlarged folds	H0	49 (98.0%)	50 (100%)	1
	H1	1 (2.0%)	0 (0%)	
Nodular gastritis	N0	50 (100%)	50 (100%)	1
	N1	0 (0%)	0 (0%)	
Diffuse redness	DR0	46 (92.0%)	46 (92.0%)	1
	DR1	4 (8.0%)	4 (8.0%)	
	DR2	0 (0%)	0 (0%)	

**Table 3 jcm-11-02198-t003:** Endoscopic severity of gastritis by white-light imaging according to the Kyoto classification of gastritis.

Endoscopic Characteristic	GIF-1200N Endoscope	GIF-290N Endoscope	*p*-Value
Atrophy	1.66 ± 0.56	1.68 ± 0.47	0.709
Intestinal metaplasia	1.16 ± 0.87	1.26 ± 0.80	0.168
Enlarged folds	0.02 ± 0.14	0.00 ± 0.00	0.322
Nodular gastritis	0.00 ± 0.00	0.00 ± 0.00	
Diffuse redness	0.08 ± 0.27	0.08 ± 0.33	1.000
Total score	2.94 ± 1.641	3.12 ± 1.29	0.060

Data show mean ± standard deviation.

**Table 4 jcm-11-02198-t004:** Color differences between areas with and without atrophy and between regions with and without intestinal metaplasia.

Finding		GIF-1200N Endoscope	GIF-290N Endoscope		GIF-1200N Endoscope	GIF-290N Endoscope	
		WLI	WLI	*p*-Value	NBI	NBI	*p*-Value
Atrophy	ΔL*	5.8 ± 11.0	5.7 ± 7.9	0.954	12.3 ± 13.0	5.6 ± 11.5	**0.030**
	Δa*	−6.0 ± 4.7	−8.8 ± 7.3	**0.015**	−1.6 ± 10.3	5.5 ± 6.5	**0.002**
	Δb*	−0.6 ± 3.0	−2.4 ± 4.1	**0.012**	2.0 ± 3.7	2.3 ± 2.9	0.733
	ΔE*	13.4 ± 6.4	15.0 ± 4.8	0.133	19.2 ± 8.5	14.4 ± 6.2	**0.001**
Intestinal metaplasia	ΔL*	2.2 ± 4.7	3.3 ± 4.9	0.034	5.0 ± 6.7	3.0 ± 8.5	0.388
	Δa*	−0.6 ± 5.3	−2.5 ± 6.0	0.076	−9.3 ± 5.8	−9.0 ± 5.0	0.819
	Δb*	−0.5 ± 2.7	−1.7 ± 3.3	**0.043**	−2.3 ± 3.6	−2.9 ± 2.0	0.415
	ΔE*	7.1 ± 3.3	6.9 ± 3.1	0.738	13.7 ± 4.2	13.3 ± 4.4	0.760

Abbreviations: WLI: white-light imaging, ΔL*: change in brightness, Δa*: change in red-green component, Δb*: change in yellow-blue component, ΔE: color difference; * *p* < 0.05. Bold meant data with *p* < 0.05 with significant different.

## Data Availability

The data presented in this study are available on request from the corresponding author.

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
