# Peer review of "Efficacy of a Third-Generation High-Vision Ultrathin Endoscope for Evaluating Gastric Atrophy and Intestinal Metaplasia in Helicobacter pylori-Eradicated Patients"

_jcm, 2022, doi:10.3390/jcm11082198_

Round 1
Reviewer 1 Report
Interesting work! I've appreciated the interest in early diagnosis of cancer / precancerous lesions. This issue is very important to be considered even in western countris. I'would like a more complete explanation about the images obtained (even just some picturs).
Author Response
Thank you for your comments. We believe that this study will be of interest to the readership of Journal of Clinical Medicine.
With agreement to your comments, we added comments about endoscopic atrophy and intestinal metaplasia in legend of Figure 1 of the revised version, as below.
Figure 1. Images taken using a second-generation GIF-290N ultrathin endoscope by white-light imaging (A), third-generation GIF-1200N ultrathin endoscope by white-light imaging (B), second-generation endoscope by narrow-band imaging (C) and third-generation endoscope by narrow-band imaging (D). The endoscopic features of mucosal atrophy are characterized by a discolored mucosa and visible capillary network in the atrophic area (A to D). Intestinal metaplasia is defined as multiple ashen nodular or cobblestone-like lesions on atrophic mucosa observed (A and B). Villous appearance, whitish mucosa, and rough mucosal surface are helpful indicators for the endoscopic diagnosis of intestinal metaplasia. Map-like redness is defined as reddish depressed areas of various shape and sizes in the atrophic area (A).
Reviewer 2 Report
This is an interesting and nicely written manuscript in which the authors compared and seek for potential differences and improvements between a 2nd and 3rd generation nasal scopes in patients that hade received h pylori eradication.
The manuscript has some issues that need to be addressed, revised and mentioned.
- Who performed the 1st and 2nd endoscopies. An how much longer after the 1st the patients had the 2nd one performed. This is important to know and report since time lead bias as well as recall bias might interfere with the findings and results. Were the endoscopist blinded to the first result
- Please elaborate more on the potential bias and modifications expected on the gastric mucosa, atrophy and severity of gastritis after erradiaciton and how this would definitely affect interpretation of results and if it might indeed be responsible of the reported enhancement.
- Second conclusion is speculative and represents more an hypothesis. If a 3rd generation nasal scope offers better gastric cancer screening performance is not supported by the results and was not an objective of the investigation. I consider this sentences a more suitable and supported 2nd conclusion "
NBI using a third-generation endoscope produced significantly 199 greater color differences surrounding atrophy and intestinal metaplasia compared with WLI"
Author Response
Our responses to comments raised by the Reviewer 2
- Who performed the 1st and 2nd endoscopies. How much longer after the 1st the patients had the 2nd one performed. This is important to know and report since time lead bias as well as recall bias might interfere with the findings and results. Were the endoscopist blinded to the first result.
Response:
Thank you for your comments. As you mentioned, endoscopist performed endoscopy and duration of the 1st and 2nd endoscopies are important factors in this study. In this study, all endoscopy was performed one expert endoscopist, KT. Therefore, we added information of endoscopist and duration of the 1st and 2nd endoscopies in the revised version, as below.
Line 103
Endoscopy was performed using the older second-generation GIF-290N (from March, 2015 to June, 2020) and new third-generation ultrathin GIF-1200N endoscopes (from April, 2021 to January, 2022) (Figure 1).
Line 109
The 1st and 2nd endoscopies were performed by an expert endoscopist (KT).
Table 1.
Mean duration form 1st to 2nd endoscopies (months ± SD): 35.5 ± 19.8
- Please elaborate more on the potential bias and modifications expected on the gastric mucosa, atrophy and severity of gastritis after eradication and how this would definitely affect interpretation of results and if it might indeed be responsible of the reported enhancement.
Response:
In general, gastric mucosal atrophy after H. pylori eradication therapy in any of patients is known to be gradually recovered depended on age, severity of atrophy at eradication and environmental and genetic factors. In this study, because mean duration form 1st to 2nd endoscopies was 35.5 ± 19.8 months, when ability for evaluation of gastritis is similar between second-generation GIF-290N and third-generation GIF-1200N endoscopes, the gastritis score by GIF-1200N endoscope may be lower than those by GIF-290N. However, mean endoscopic scores of gastritis evaluated using WLI according to the Kyoto classification of gastritis were not significantly different between GIF-290N and GIF-1200N endoscopes in this study. Therefore, although ideally, two endoscopies should be done at the same time, difference of gastric mucosal situation by duration form 1st to 2nd endoscopies may ignore. However, as you suggested, because this study may have potential bias and this is serious limitation of this study, we added comments as limitation in the revised version, as below.
Line 313
Endoscopic evaluations of gastritis using GIF-290N and GIF-1200N were performed at different times. In general, gastric atrophy and intestinal metaplasia after eradication therapy in any of patients is gradually recovered depended on age, severity of atrophy and intestinal metaplasia at eradication and environmental and genetic factors. In this study, because mean duration form 1st to 2nd endoscopies was 35.5 ± 19.8 months, the severity of gastritis may differ between GIF-290N andGIF-1200N endoscopes. Because two endoscopies should be done at the same time to compare ability, this study may have potential bias. However, mean endoscopic scores of gastritis evaluated using WLI according to the Kyoto classification of gastritis were not significantly different between GIF-290N and GIF-1200N endoscopes, difference of gastric mucosal situation by duration form 1st to 2nd endoscopies may ignore. However, we will plan to clarify the clinical endoscopic significance between GIF-290N and GIF-1200N at the same time, as the further study.
- Second conclusion is speculative and represents more a hypothesis. If a 3rd generation nasal scope offers better gastric cancer screening performance is not supported by the results and was not an objective of the investigation. I consider this sentence a more suitable and supported 2nd conclusion" NBI using a third-generation endoscope produced significantly greater color differences surrounding atrophy and intestinal metaplasia compared with WLI"
Response:
Thank you for your suggestion and we agreed with your comments.
We revised the Conclusion of Abstract in “NBI using a third-generation ultrathin endoscope produced significantly greater color differences surrounding atrophy and intestinal metaplasia in H. pylori-eradicated patients compared with WLI.” in the revised version.
Reviewer 3 Report
Reviewer’s Comments
Summary and general comments:
In the current retrospectively study, the authors randomly selected 50 Helicobacter pylori-eradicated patients who had previously received endoscopic examination with a second-generation endoscope and later underwent examination with a third-generation endoscope and compared the color difference between the images from the same patient using two imaging techniques [i.e., narrow-band imaging (NBI) and white-light imaging (WLI)]. The authors found that the color differences surrounding atrophy produced by NBI using the third-generation endoscope were significantly greater than those using the second-generation endoscope but no notable color differences surrounding intestinal metaplasia between WLI and NBI. Therefore, the authors proposed that combining NBI with a third-generation endoscope may be useful for enhancing color differences for detecting atrophic borders and identifying gastric cancer. Despite the potential clinical significance of the results, potential changes in disease status during the time interval between the two examinations and a lack of clinical correlation to support the significance of the authors’ findings are the two weaknesses of the current study as detailed below.
Major comments
- Based on the authors’ description about this retrospective study in section 2.1 and the limitation section, it was apparent that the two examinations (i.e., first exam using second-generation endoscope and second exam with third-generation endoscope) were not conducted at the same time. What were the time intervals between the second-generation endoscopic exam and the third-generation exam for the 50 patients?
- In continuation with the question above, the major concern is the possible changes in disease (e.g., gastritis) severity during the interval between the first and second endoscopic examinations because the comparison of five parameters (i.e., atrophy, intestinal metaplasia, hypertrophy of gastric folds, nodularity, and diffuse redness) can be justified only on the assumption that there were no notable pathological changes between the two examinations. This assumption, however, may not be correct, especially when there was a long interval between the two examinations in an aged population (mean age: 74.3 ± 6.6).
- Usually when two methods are compared in terms of their clinical merits, they need to be compared with a gold standard. Because the five parameters for comparison (i.e., atrophy, intestinal metaplasia, hypertrophy of gastric folds, nodularity, and diffuse redness) were not pathologically supported in the current study, what were the gold standards based on which the two generations of endoscopes (or imaging techniques) were compared? In other words, the apparently better image quality (i.e., greater color difference) of a particular imaging technique (or generation of endoscope) for an outcome (e.g., atrophy or intestinal metaplasia) may actually be an artifact during the image management process (e.g., Photoshop in this study).
- The authors mentioned in section 2.3: “We randomly set three pairs (with and without) of regions (atrophy or intestinal metaplasia) of interest and calculated the color difference in each patient.” If the regions were randomly chosen without reference to any anatomical landmarks, how did the authors ensure that the three pairs of regions were identical for comparison in both sets of images (i.e., one from second-generation and the other from third generation endoscope)?
Minor comments
The authors described in section 2.1: “This study was a retrospective trial … in patients who had received eradication therapy for H. pylori infection.’ However, the authors mentioned at the end of the same section describing the exclusion criteria: “Exclusion criteria were patients with a history of … and a history of eradication therapy for H. pylori infection.” Based on the descriptions, it is unclear whether the authors actually included or excluded patients who had received eradication therapy for H. pylori infection.
Author Response
Our responses to comments raised by the Reviewer 3
Major comments
- Based on the authors’ description about this retrospective study in section 2.1 and the limitation section, it was apparent that the two examinations (i.e., first exam using second-generation endoscope and second exam with third-generation endoscope) were not conducted at the same time. What were the time intervals between the second-generation endoscopic exam and the third-generation exam for the 50 patients?
Response:
In this study, as you suggested, the two examinations were not conducted at the same time and mean duration form 1st to 2nd endoscopies was 35.5 ± 19.8 months. We added information of duration of the 1st and 2nd endoscopies in the revised version, as below.
Line 103
Endoscopy was performed using the older second-generation GIF-290N (from March, 2015 to June, 2020) and new third-generation ultrathin GIF-1200N endoscopes (from April, 2021 to January, 2022).
Table 1.
Mean duration form 1st to 2nd endoscopies (months ± SD): 35.5 ± 19.8
- In continuation with the question above, the major concern is the possible changes in disease (e.g., gastritis) severity during the interval between the first and second endoscopic examinations because the comparison of five parameters (i.e., atrophy, intestinal metaplasia, hypertrophy of gastric folds, nodularity, and diffuse redness) can be justified only on the assumption that there were no notable pathological changes between the two examinations. This assumption, however, may not be correct, especially when there was a long interval between the two examinations in an aged population (mean age: 74.3 ± 6.6).
Response:
We agreed with your comments. In general, gastric atrophy and intestinal metaplasia after H. pylori eradication therapy in any of patients is known to be gradually recovered depended on age, severity of atrophy and intestinal metaplasia at eradication and environmental and genetic factors. In this study, because mean duration form 1st to 2nd endoscopies was 35.5 ± 19.8 months, it cannot be denied that the severity of gastritis may have changed at both points in time. Although pathological examination is considered as gold standard for gastric atrophy and intestinal metaplasia in the world, we did not evaluate pathological atrophy and intestinal metaplasia at both points. This is limitation of this study.
When ability for evaluation of gastritis is similar between second-generation GIF-290N and third-generation GIF-1200N endoscopes, the gastritis score by GIF-1200N endoscope may be lower than those by GIF-290N. However, mean endoscopic scores of gastritis evaluated using WLI according to the Kyoto classification of gastritis were not significantly different between GIF-290N and GIF-1200N endoscopes in this study. Therefore, although ideally, two endoscopies should be done at the same time, difference of gastric mucosal situation by duration form 1st to 2nd endoscopies may ignore.
However, as you suggested, because this is serious limitation of this study, we added comments as limitation in the revised version, as below.
Line 313
Endoscopic evaluations of gastritis using GIF-290N and GIF-1200N were performed at different times. In general, gastric atrophy and intestinal metaplasia after eradication therapy in any of patients is gradually recovered depended on age, severity of atrophy and intestinal metaplasia at eradication and environmental and genetic factors. In this study, because mean duration form 1st to 2nd endoscopies was 35.5 ± 19.8 months, the severity of gastritis may differ between GIF-290N andGIF-1200N endoscopes. Because two endoscopies should be done at the same time to compare ability, this study may have potential bias. However, mean endoscopic scores of gastritis evaluated using WLI according to the Kyoto classification of gastritis were not significantly different between GIF-290N and GIF-1200N endoscopes, difference of gastric mucosal situation by duration form 1st to 2nd endoscopies may ignore.
- Usually when two methods are compared in terms of their clinical merits, they need to be compared with a gold standard. Because the five parameters for comparison (i.e., atrophy, intestinal metaplasia, hypertrophy of gastric folds, nodularity, and diffuse redness) were not pathologically supported in the current study, what were the gold standards based on which the two generations of endoscopes (or imaging techniques) were compared? In other words, the apparently better image quality (i.e., greater color difference) of a particular imaging technique (or generation of endoscope) for an outcome (e.g., atrophy or intestinal metaplasia) may actually be an artifact during the image management process (e.g., Photoshop in this study).
Response:
Thank you for your comments. We also agreed that when two methods are compared in terms of their clinical merits, they need to be compared with a gold standard. Endoscopic quality generally depends on endoscopist, endoscopic instrumentation, kinds of image-enhancement techniques and preparation before endoscopy. In this study, to avoid bias of endoscopist, all endoscopy was performed one expert endoscopist, KT.
Although pathological examination is considered as gold standard for evaluation of gastric atrophy and intestinal metaplasia, because we did not evaluate pathological gastritis, we had no gold standard in the two generations of endoscopes. Therefore, as you suggested, the apparently better image quality of a particular imaging technique for an evaluation of atrophy and intestinal metaplasia may actually be an artifact during the image management process. Therefore, we added this point in the revised version, as limitation, as below.
Line 109
The 1st and 2nd endoscopies were performed by an expert endoscopist (KT).
Line 328
Forth, although pathological examination is considered as gold standard for evaluation of gastric atrophy and intestinal metaplasia, we did not evaluate pathological gastritis in this study. Therefore, we cannot completely deny a hypothesis that the apparently better image quality of a particular imaging technique for an evaluation of atrophy and intestinal metaplasia may actually be an artifact during the image management process.
- The authors mentioned insection 2.3: “We randomly set three pairs (with and without) of regions (atrophy or intestinal metaplasia) of interest and calculated the color difference in each patient.” If the regions were randomly chosen without reference to any anatomical landmarks, how did the authors ensure that the three pairs of regions were identical for comparison in both sets of images (i.e., one from second-generation and the other from third generation endoscope)?
Response:
In this study, we randomly set three pairs (with and without) of regions (atrophy or intestinal metaplasia) of interest and calculated the color difference in each patient using pictures of similar anatomical location (i.e., antrum and lesser curve of lower body) at both GIF-290N and GIF-1200N endoscopes.
We have revised the description so that it is not misunderstood in the revised version.
Line 125
We randomly set three pairs (with and without) of regions (atrophy or intestinal metaplasia) of interest and calculated the color difference in each patient using pictures of similar anatomical location (i.e., antrum and lesser curve of lower body) at both GIF-290N and GIF-1200N endoscopes.
Minor comments
- The authors described insection 2.1: “This study was a retrospective trial … in patients who had received eradication therapy for pylori infection.’ However, the authors mentioned at the end of the same section describing the exclusion criteria: “Exclusion criteria were patients with a history of … and a history of eradication therapy for H. pylori infection.” Based on the descriptions, it is unclear whether the authors actually included or excluded patients who had received eradication therapy for H. pylori infection.
Response:
Thank you for your comment. This is our mistake. We revised this part as “no history of eradication therapy for H. pyloriinfection” in the revised version (Line 94).
Round 2
Reviewer 3 Report
Despite the limitations of the present study, the authors have adequately addressed them and reinforced relevant descriptions in their revised version. I have no further questions. Nevertheless, some grammatical mistakes and misspellings need to be taken care of. For example, "duration form" should be "duration from", "to ignore" should be "to be ignored", and "Forth" should be "Fourth", just to name a few.
Author Response
Our responses to comments raised by the Reviewer 3
- Despite the limitations of the present study, the authors have adequately addressed them and reinforced relevant descriptions in their revised version. I have no further questions. Nevertheless, some grammatical mistakes and misspellings need to be taken care of. For example, "duration form" should be "duration from", "to ignore" should be "to be ignored", and "Forth" should be "Fourth", just to name a few.
Response:
Thank you for your comments. As you recommended, we carefully revised some grammatical mistakes and misspellings in the revised version.